# CTSF: An Intrusion Detection Framework for Industrial Internet Based on Enhanced Feature Extraction and Decision Optimization Approach

**DOI:** 10.3390/s23218793

**Published:** 2023-10-28

**Authors:** Guangzhao Chai, Shiming Li, Yu Yang, Guohui Zhou, Yuhe Wang

**Affiliations:** College of Computer Science and Information Engineering, Harbin Normal University, Harbin 150025, China; cgz010321@163.com (G.C.); yang11zc@163.com (Y.Y.); zhouguohui@hrbnu.edu.cn (G.Z.)

**Keywords:** Industrial Internet, intrusion detection, convolutional neural network, Transformer, Support Vector Machines

## Abstract

The traditional Transformer model primarily employs a self-attention mechanism to capture global feature relationships, potentially overlooking local relationships within sequences and thus affecting the modeling capability of local features. For Support Vector Machine (SVM), it often requires the joint use of feature selection algorithms or model optimization methods to achieve maximum classification accuracy. Addressing the issues in both models, this paper introduces a novel network framework, CTSF, specifically designed for Industrial Internet intrusion detection. CTSF effectively addresses the limitations of traditional Transformers in extracting local features while compensating for the weaknesses of SVM. The framework comprises a pre-training component and a decision-making component. The pre-training section consists of both CNN and an enhanced Transformer, designed to capture both local and global features from input data while reducing data feature dimensions. The improved Transformer simultaneously decreases certain training parameters within CTSF, making it more suitable for the Industrial Internet environment. The classification section is composed of SVM, which receives initial classification data from the pre-training phase and determines the optimal decision boundary. The proposed framework is evaluated on an imbalanced subset of the X-IIOTID dataset, which represent Industrial Internet data. Experimental results demonstrate that with SVM using both “linear” and “rbf” kernel functions, CTSF achieves an overall accuracy of 0.98875 and effectively discriminates minor classes, showcasing the superiority of this framework.

## 1. Introduction

Industrial Internet [1] is an emerging domain that amalgamates conventional industrial systems with internet technologies. Leveraging advanced techniques like the Internet of Things (IoT), cloud computing, and big data analytics, it facilitates the interconnectedness and intelligent management of large-scale devices within industrial production environments. The proliferation of Industrial Internet and its digital transformation has brought forth numerous challenges [2]. Firstly, as the networking degree of industrial control systems escalates, industrial networks become exposed to a plethora of threats. Malicious actors can disrupt, interfere, or even gain control over industrial systems through network intrusion and malevolent activities, severely impacting industrial production and infrastructure. Secondly, security vulnerabilities and weaknesses in the Industrial Internet have gradually become targets for attackers. Industrial control systems often employ antiquated devices and software, lacking timely security patches and updates. Concurrently, attackers exploit advanced network attack techniques such as zero-day vulnerabilities and targeted assaults on specific industrial protocols, infiltrating and penetrating the Industrial Internet. Lastly, the Industrial Internet harbors a substantial amount of sensitive production data and operational information. Should these data fall into the hands of malicious entities, it can lead to substantial losses for enterprises. Furthermore, malicious intrusions can result in severe consequences including production interruptions, equipment damage, and industrial accidents, thereby posing significant threats to the production and economic interests of enterprises.

To bolster the security of Industrial Internet systems, traditional network security methods are adapted for use in Industrial Internet contexts. Figure 1 depicts a generalized security architecture for safeguarding Industrial Internet platforms. The left portion of the diagram represents the entire lifecycle management process of data in the Industrial Internet, with a higher susceptibility to malicious attacks during the data collection and transmission phases. In the middle section of the diagram, passive defense measures against malicious attacks are presented, including the common service models in cloud computing: SaaS, PaaS, and IaaS. These models play a pivotal role in establishing and providing secure computing and storage environments. On the right side of the diagram, active defense measures against malicious attacks are depicted; the Intrusion Detection System (IDS) represents an active defense component within the depicted framework, playing a significant role in the Industrial Internet context. IDS, a form of network security tool [3,4], is designed to monitor and identify potential intrusion activities, thereby safeguarding the integrity of Industrial Internet systems. By implementing effective IDS solutions, the Industrial Internet can ensure the reliability, integrity, and confidentiality of its systems, thereby establishing a stable and secure environment for industrial production. IDS can be categorized into signature-based IDS (SIDS) and anomaly-based IDS (AIDS) [5]. SIDS relies on predefined attack patterns or feature databases for intrusion detection, performing matching against network traffic, system logs, or other data using known attack pattern signatures. However, it can only identify known intrusions. On the other hand, AIDS is a method that detects abnormal activities by analyzing the normal behavioral patterns of a system. AIDS constructs baseline models of normal network traffic, system behavior, and user activities to detect anomalous behavior that deviates from the established normal model. This approach can identify unknown attacks and emerging threats.

In the context of AIDS, machine learning techniques (ML) are extensively employed to enhance the automation and intelligence of IDS, thereby reducing false positive rates and better addressing evolving network threats [6,7,8]. Support Vector Machine (SVM), a classical ML algorithm, finds widespread application in intrusion detection [9,10]. SVM boasts significant advantages in handling high-dimensional data, generalization, small-sample data, and linear and non-linear problems. However, in IDS, data features can sometimes be redundant, or their impact on accurately classifying data points might be marginal. Hence, SVM models are often used in conjunction with feature selection algorithms and model optimization methods to maximize SVM’s classification accuracy [11]. An effective feature selection approach involves the use of deep learning networks (DL) [12,13,14]. DL, achieved through the combination and training of multi-layered neural networks, can automatically acquire sophisticated abstract feature representations from raw data, greatly streamlining the feature extraction process. At its core, DL utilizes neural network models [15], comprising multiple layers of neurons, with each layer performing a series of non-linear transformations and feature extractions on input data. As the network depth increases, DL gradually learns more abstract and complex feature representations, thereby achieving a deep understanding and analysis of the data. Transfomer, a DL network architecture based on self-attention mechanisms [16], dynamically computes attention weights for each position based on the relationships between different feature positions in the input data. By summing the weighted attention weights for all positions, the Transformer model can simultaneously consider the feature information of the entire input data, thus capturing global feature relationships more effectively. However, when it comes to local relationships within input data, its modeling capacity is limited.

The self-attention mechanism calculates attention weights by considering all positions in the sequence, making it challenging to focus attention effectively on locally related positions. Capturing local features is crucial for Industrial Internet intrusion detection. The network traffic and data in Industrial Internet environments typically contain a wealth of subtle local information; this information may appear in specific segments or time intervals within the entire data stream, such as specific behaviors, patterns, or features associated with attacks. Capturing these local features enables more accurate identification of anomalies or potential intrusion behaviors. Furthermore, focusing solely on global features can make the model more susceptible to interference from normal variations or noise, leading to false alarms. By capturing local features, the model can better distinguish between normal and abnormal changes, thus reducing the false positive rate. Shallow convolutional kernels of convolutional neural networks (CNN) [17,18,19] precisely extract local features from data and leverage parameter sharing mechanisms to significantly reduce the network’s parameter count.

Therefore, the CNN-Transformer network structure not only automatically extracts features but also compensates for the Transformer model’s deficiencies in local feature extraction by harnessing CNN’s precise extraction of local features [20].

In consideration of various factors, this paper presents a network framework named CTSF for intrusion detection in the Industrial Internet. CTSF’s pre-training phase initially submits input data to a CNN for extracting sufficient local features. Subsequently, these features undergo positional encoding and are passed to an enhanced Transformer’s encoder structure. The self-attention mechanism is employed for holistic feature extraction. Finally, the decoder, composed of fully connected layers, further abstracts the encoded features. After ample training in the pre-training phase, CTSF extracts the final layer’s output from this phase, which is then passed to the SVM in the decision-making segment for further training, ultimately yielding classification outcomes. In theory, due to CTSF addressing the limitations of traditional Transformer and SVM models, its classification accuracy is expected to be higher than the unimproved models it comprises. This proposition will also be substantiated in the subsequent experimental section. The decoder of CTSF’s pre-training phase consists of fully connected layers and does not receive label data or employ self-attention mechanisms for label-specific feature extraction. This is due to the fact that in the context of abnormal traffic in the Industrial Internet, label data often represent a category of attack and possess only one-dimensional features. While utilizing self-attention mechanisms to extract features from label data might yield marginal effects, it significantly increases the overall network parameters. Figure 2 illustrates the fundamental process of the traditional Transformer in handling data with high-dimensional label classes, as well as the modified version proposed in this paper for processing Industrial Internet network traffic data.

Introduces a novel network framework, CTSF, tailored for intrusion detection in the Industrial Internet. This framework is rooted in the actual conditions of the Industrial Internet environment and modifies the traditional Transformer model’s decoder structure to better handle abnormal traffic.The CTSF framework leverages the advantage of CNN in extracting local features, compensating for the traditional Transformer’s shortcomings in this aspect, and effectively applying to intrusion detection in the Industrial Internet.This paper conducts simulation experiments on the Industrial Internet dataset X-IIOTID with CTSF, and the results demonstrate CTSF’s ability to accurately recognize small-sample categories, achieving an overall accuracy of 0.98875.

The remaining sections of this paper are organized as follows: Section 2 discusses the relevant background; Section 3 provides a comprehensive description of CTSF’s structure, forward propagation, and training process; Section 4 introduces the selected dataset and its processing methodology, model selection for comparative experiments, specifications of each model’s parameters, model evaluation metrics, and final experimental outcomes; Section 5 concludes the paper and proposes future avenues for improvement.

## 2. Related Work

### 2.1. SVM

SVM, a classical method in the field of machine learning, has been widely applied and has shown remarkable performance in intrusion detection [21]. With its unique characteristics and advantages, SVM has become a powerful tool for addressing classification and regression problems. SVM typically requires the use of feature selection algorithms to achieve its maximum effectiveness [11]. Almaiah et al. introduced an intrusion detection model employing SVM with feature selection using principal component analysis and various kernel functions [22]. They examined the influence of different kernel functions on SVM and found that SVM combined with the ‘rbf’ kernel function and principal component analysis feature selection technique achieved the highest accuracy. Saheed et al. attempted to address the dynamics of network attack patterns using the k-means clustering algorithm, followed by feature selection using genetic algorithms, and finally employed SVM for classification [23]. This approach achieved an impressive accuracy of 99% at low FAR, showcasing remarkable success. Not only in the realm of feature selection, but in other domains as well, researchers have utilized CNN for rapid feature extraction and employed the extracted features as input data for SVM classification. For instance, in the field of medical tumor detection, Khairandish et al. combined CNN with SVM [24], harnessing the automatic feature extraction capabilities of CNN and the high accuracy and speed of SVM to successfully distinguish between benign and malignant brain tumors. In the domain of weed recognition, Tao et al. used CNN to extract features from crop images [25], eliminating the need for manual feature extraction and demonstrating that CNN-SVM achieves higher classification accuracy and robustness.

### 2.2. CNN

CNN finds extensive application in intrusion detection within the Industrial Internet, automatically learning and extracting features from input data [26,27]. Utilizing convolutional and pooling layers, CNN can effectively capture both local and global patterns in sequential data, encompassing crucial feature correlations and dependencies. These learned features can be employed to identify potential intrusion behaviors and anomalous activities. Pingale S V and his team have proposed a robust Remora Whale Optimization (RWO)-based hybrid deep model [28]. This model preprocesses data, extracts features using CNN, and then employs a hybrid algorithm for training, achieving outstanding classification performance. El-Ghamry et al. employed various CNN architectures [29], such as VGG16, Inception, and Xception, to further abstract and process network traffic converted into color images. They also compared the performance of these CNN architectures with other machine learning algorithms, ultimately demonstrating the performance advantage and high classification accuracy of CNN architectures. Qazi et al. proposed a deep learning architecture based on one-dimensional convolutional neural networks [30], which maintained structural simplicity and low computational cost while exhibiting remarkably high classification accuracy. Halbouni et al. developed an intrusion detection system using a CNN-based learning algorithm [31], empirically confirming that increasing the number of layers and neurons per layer can enhance model performance. This intrusion detection system achieved a 99.55% accuracy and 99.63% detection rate in multi-class classification. Despite CNN’s remarkable performance in intrusion detection, its effectiveness in global feature extraction is limited by the receptive field range [32]. To overcome this limitation, some researchers have begun combining CNN with models that excel in extracting global features, and one such model to be mentioned next is the Transformer.

### 2.3. Transformer

Transformer, a neural network model based on self-attention mechanisms, has been widely employed in the field of natural language processing. However, its advantages in sequence modeling and global feature extraction can also be applied to the domain of intrusion detection. Wu et al. introduced a Robust Transformer-based Intrusion Detection System (RTIDS) [33], which employs positional embedding techniques to correlate sequential information among features. Through a variant of stacked encoder–decoder architecture, it learns low-dimensional feature representations from high-dimensional raw data. Experimental comparisons with neural networks such as RNNs and LSTMs that extract sequence information validate the effectiveness of this Transformer-based intrusion detection system.

Wang et al. proposed a Transformer-based NIDS method for IoT [34], utilizing the self-attention mechanism in encoders and decoders to learn contextual relationships among input network features, resulting in outstanding classification performance on the ToN-LoT dataset. Liu et al. introduced an enhanced intrusion detection model based on Transformer [35], primarily altering the positional encoding method of the traditional Transformer model. By embedding the positional information of features, they effectively captured feature dependencies, consequently improving classification accuracy. Tan et al. presented a Transformer-based intrusion detection system [36] which, due to the absence of recurrence and the use of slot-based functions, can achieve real-time detection efficiency exceeding that of bidirectional LSTMs.

While Transformer exhibits strong capabilities in global feature extraction in the aforementioned studies, it still has limitations when applied to Industrial Internet of Things intrusion detection. Firstly, it lacks an explicit mechanism for local modeling. The Transformer model primarily models relationships between elements in a sequence using self-attention mechanisms, without a dedicated mechanism for explicit handling of local features. This implies that the Transformer model might rely on global context when processing local information, potentially leading to imprecise capture of local features. Secondly, in industrial IoT environments, network traffic labels often consist of only one dimension. In the traditional Transformer architecture, the decoder employs self-attention mechanisms to process label data. When network traffic is passed into the Transformer architecture, the decoder expends significant time and parameters to extract features from one-dimensional label data, which is undesirable.

Addressing the aforementioned issues, one solution for the first problem is to combine the advantages of CNN-based local feature extraction and Transformer-based global feature extraction, thereby compensating for the deficiencies of both architectures. A solution to the second problem is proposed in the third section of this paper.

### 2.4. CNN-Transformer

The CNN-Transformer fusion combines the advantages of both the CNN and Transformer model, simultaneously possessing the capabilities of local feature extraction and global context modeling [37]. Yao R et al. introduced a hybrid intrusion detection system architecture based on CNN-Transformer [38]. They first employed an XGBoost-based feature selection strategy to eliminate irrelevant features from network traffic. Subsequently, they constructed a CNN to extract local features from the traffic and then utilized a Transformer to establish feature correlations and extract global features. The effectiveness and superiority of this hybrid structure were demonstrated through testing on the KDDCup99, NSL-KDD, and CICIDS-2017 datasets. Luo S et al. proposed a layered intrusion detection model with spatio-temporal feature fusion based on CNN-Transformer [39]. They leveraged CNN to extract spatial features and employed the Transformer’s positional encoder to embed location information into input data, cleverly capturing time features while globally modeling data. This model highlights the significance of the Transformer’s attention mechanism in soft feature selection. Their experiments on the UNSW-NB15 dataset underscore the advancement of the proposed model. Indeed, what sets CTSF apart from the models proposed by the aforementioned researchers is the incorporation of SVM as a new final decision-making component. With the inclusion of SVM, CTSF combines the strengths of both the softmax classifier and the SVM classifier, thus further enhancing classification accuracy.

## 3. Framework Analysis

In the field of industrial production and operations, specific types of equipment, sensors, and control systems generate network traffic data within the Industrial Internet environment. This data serves to record real-time status, performance parameters, and operational conditions of industrial processes, facilitating remote monitoring, data analysis, and automated control. These devices encompass a range of sensors (such as temperature, humidity, pressure, and flow sensors), PLCs (Programmable Logic Controllers), SCADA systems (Supervisory Control and Data Acquisition systems), industrial robots, IoT devices, data acquisition cards, industrial switches, routers, and more. Through these devices, the Industrial Internet environment can produce diverse network traffic data to support the efficient management and optimization of industrial processes. This data contains various information crucial for achieving intelligent production, monitoring, and management. To extract features more accurately from network traffic for enhanced classification precision, a novel intrusion detection framework named CTSF is proposed in this paper. It comprises two components: a pre-training section and a decision-making section. The pre-training section of CTSF is composed of a CNN and an improved Transformer structure tailored to the domain of Industrial Internet intrusion detection, while the classification section involves an SVM classifier. The overall structure is illustrated in Figure 3. In this section, we will derive the forward propagation process of CTSF, present the parameters utilized during this process, and elucidate their respective significance as shown in Table 1. The parameters in Table 1 represent abstract symbols, and these symbols collaborate during the inference process of forward propagation, mathematically illustrating the feature extraction and model prediction processes of CTSF.

### 3.1. Pre-Training Part

#### 3.1.1. CNN in the Pre-Training Part

In the pre-training phase, CTSF primarily employs CNN for extracting local features from the input data. Assume a network traffic event X with a shape of (1, n), where n represents the number of features in X. A convolutional kernel is used with a shape of $\left(1,N_{c}\right)$ and a movement stride of $step_{c}$. Considering that the convolution operation reduces the dimension of the original feature vector, CTSF applies zero-padding to both ends of the data before convolution.

This step aims to minimize the potential loss of edge features. The padding process is demonstrated by Formula (1):(1)$$X=[0,x_{1},x_{2},x_{3},...,x_{n},0]$$

After the padding operation, the shape of X becomes (1, n + 2). Following the padding, CTSF proceeds with the convolution operation. Each initial feature vector at every stride undergoes convolution to yield the process value $X_{c\_i}$. This process is illustrated by Formula (2):(2)$$X_{c\_i}=ReLU(w_{c}X_{i:i+N_{c}}^{T}+b_{c})$$

Here, $w_{c}$ and $b_{c}$ represent the weight parameters and bias vector of the convolutional kernel, respectively. After completing all strides for each convolutional kernel, CTSF obtains $X_{c}$, with a shape of $\left(1,N_{p}\right)$, where $N_{p}$ is computed as demonstrated in Formula (3). In fact, $N_{p}$ is numerically equal to n, as is due to the padding operation performed by CTSF before the convolution.
(3)$$N_{p}=\left\lceil \frac{n-N_{c}+3}{step_{c}}\right\rceil$$

Subsequently, $X_{c}$ will undergo a max-pooling layer with a shape of (1, $N_{m}$) and a stride of $step_{m}$. The process of obtaining $X_{m}$ is illustrated in Formulas (4)–(6):(4)$$X_{m\_i}=Max(X_{c\_i},X_{c\_i+1},...,X_{c\_i+N_{m}-1},...,X_{c\_i+N_{m}})$$
(5)$${Num}_{m}=\left\lceil \frac{n-N_{m}+1}{step_{m}}\right\rceil$$
(6)$$X_{m}=[X_{m\_1},X_{m\_2},...,X_{m\_{Num}_{m}-1},X_{m\_Num_{m}}]$$

After undergoing multiple layers of identical convolutional operations, X results in $X_{M}$, and the mapping relationship is depicted in Formula (7):(7)$$X_{M}=f_{cnn}(X)$$

#### 3.1.2. Transformer in the Pre-Training Part

The encoder section of the pre-training phase’s Transformer structure consists of two encoder layers, each containing multi-head self-attention mechanisms, normalization layers, and feed-forward fully connected layers with residual connections. The primary purpose of the encoder is to capture the global features within the input data $X_{M}$. With the application of the self-attention mechanism, the encoder can effectively model $X_{M}$, capturing both global and selective local feature information. The decoder section comprises a fully connected layer with an output dimension matching the number of output channels from the CNN in the pre-training phase. The improved decoder part eliminates the self-attention mechanism and avoids extracting information from the labels. Instead, it transforms the features extracted from the encoder into a higher-level feature representation, reducing the training parameters in CTSF while better understanding abstract features within the input data. The structure of the pre-training phase’s Transformer is illustrated in Figure 4.

This section receives the feature information $X_{M}$ passed from the CNN section and first processes it with positional encoding. Formulas (8)–(11) illustrate the process through which CTSF obtains $X_{t}$.
(8)$${{\vec{p}}_{i}}^{\left(a\right)}=f(i{)}^{\left(a\right)}:=\left\{\begin{array}{l} \sin(\omega_{k}\cdot i),\;\;if\;a=2k\\ \cos(\omega_{k}\cdot i),\;\;if\;a=2k+1 \end{array}\right.$$
(9)$$\omega_{k}=\frac{1}{10000^{2k/d_{\mathrm{model}}}}$$
(10)$$X_{t\_i}=X_{M\_i}+{\vec{p}}_{i}$$
(11)$$X_{t}=[X_{t\_1},X_{t\_2,...,}X_{t\_num}]$$

Here, a represents an integer value that controls the odd and even conditions in ${\vec{p}}_{i}$, num which is equal to the feature dimension of $X_{M}$, and $d_{\mathrm{model}}$ represents the dimension by which each feature is deepened.

In the multi-head attention layer of the encoder part, each feature in $X_{t}$ has three attributes: $Q=X_{t}\cdot W_{Q}$, $K=X_{t}\cdot W_{K}$, $V=X_{t}\cdot W_{V}$, where $W_{Q}$, $W_{K}$, and $W_{V}$ are learnable weight matrices. Features and attributes in $X_{t}$ are evenly mapped into multiple subspaces where the following is true: the query matrix Q maps the input sequence to the query subspace, resulting in query vectors $Q_{1},Q_{2},Q_{3}...Q_{numh-1},Q_{numh}$; the key matrix K maps the input sequence to the key subspace, resulting in key vectors $K_{1},K_{2},K_{3}...K_{numh-1},K_{numh}$; and the value matrix V maps the input sequence to the value subspace, resulting in value vectors $V_{1},V_{2},V_{3}...V_{numh-1},V_{numh}$. Here, numh represents the number of heads in the multi-head attention mechanism. Next, CTSF calculates the attention weights for each head. For the *i*-th head, CTSF computes the similarity between the query vector $Q_{i}$ and the key vector $K_{i}$, scales the similarity by the dimension $d_{K}$ of $K_{i}$, normalizes the similarity using the softmax function, and finally computes the weighted average of values using the attention weights to obtain the output $A_{i}$ of the self-attention mechanism (Formula (12)).
(12)$$A_{i}=Attention(Q_{i},K_{i},V_{i})=softmax(\frac{Q_{i}{K_{i}}^{T}}{\sqrt{d_{K}}})V_{i}$$

CTSF combines the outputs of the self-attention mechanism from all heads to obtain $X_{A}$ (Formula (13)). At this point, the encoder part has adequately captured both the global and partial local features of $X_{M}$.
(13)$$X_{A}=[A_{1},A_{2},A_{3},...,A_{numh-1},A_{numh}]$$

Formula (14) shows the process of obtaining $X_{NA}$ from $X_{A}$ through the normalization layer:(14)$$X_{NA}=\frac{X_{A}-\mu }{\sigma }$$

Here, μ represents the mean of $X_{A}$, and σ represents the standard deviation of $X_{A}$. CTSF establishes residual connections in the multi-head attention layer, as demonstrated in Formula (15).
(15)$$X_{R1}=X_{t}+F_{R1}(X_{t})$$

Here, $F_{R1}$ represents the mapping process from $X_{t}$ to $X_{NA}$. Subsequently, $X_{R1}$ undergoes a feed-forward fully connected layer and a normalization layer. Moreover, CTSF establishes residual connections in the feed-forward fully connected layer, as manifested in Formula (16):(16)$$X_{R2}=X_{R1}+F_{R2}(X_{R1})$$

In this context, $F_{R2}$ denotes the mapping process of $X_{R1}$ accomplished through the fully connected layer and the normalization layer. Subsequently, CTSF forwards X to the decoder, where it undergoes further abstraction through the decoder’s fully connected layers and partial feature dimension reduction (with CTSF’s decoder output channels being fewer than the decoder input channels), thereby reducing model training parameters. The results are then passed to the fully connected layer outside the decoder, followed by classification using the softmax function. (Formula (17))
(17)$$cls=softmax(F_{l}(X_{R2}))$$

Here, $F_{l}$ represents the mapping process of $X_{R2}$ through the fully connected layer.

### 3.2. Decision-Making Part

The CTSF framework reassigns well-pretrained initialization classification data to SVM for retraining. This design first combines the classification results of softmax and SVM to further enhance accuracy. Secondly, SVM maps this data to a high-dimensional space using a kernel function and optimizes model parameters with the hinge loss function. It searches for an optimal hyperplane in the feature space that effectively separates different class samples while maximizing the margin between the hyperplane and the nearest sample points to improve the framework’s generalization capability. After training, the SVM produces a score vector for each class using the decision function F(P), where P represents the output from the trained pre-training section: $F(P)=sign(w_{l}P+b_{l})$. Here, $w_{l}$ and $b_{l}$ represent weight parameters and bias values, respectively. introducing the argmax function. This will yield the predicted result, denoted as pred. Formula (18) shows this process:(18)pred=argmax⁡(F(P))

### 3.3. The Training Process of the CTSF

The training of CTSF consists of two parts: the initial training of the pre-training section and the final training of the decision section. Table 2 presents the overall training process of CTSF in the form of pseudocode:

## 4. Experimental Setup and Discussion

In this study, the X-IIoTID dataset from the Industrial Internet domain was employed to evaluate various performance metrics of the CTSF framework. CNN, RNN, CNN-RNN, and CNN-Transformer (improved) models were selected as control groups. These currently advanced models were also tested using the same dataset with consistent baseline parameters to derive performance metrics. Finally, a comparative analysis was conducted between the mentioned models and the CTSF framework to demonstrate the superiority of our proposed approach. Section 4.1 presents the basic information about the dataset and its preprocessing procedures. Section 4.2 outlines the basic parameter settings for the CTSF model and the control group models. Section 4.3 introduces the evaluation metrics for the models. Section 4.4 presents the analysis and discussion of the experimental results.

### 4.1. Dataset Description and Preprocessing

The X-IIoTID datasets, specifically intrusion datasets agnostic to connections and devices [40,41,42,43], were used to capture the heterogeneity and interoperability of IoT systems. These datasets encompass behaviors of emerging IoT connection protocols, recent device activities, diverse attack types and scenarios, and various attack protocols. They define attack classes and incorporate multi-view attributes including network traffic, host resources, logs, and alerts. In this study, we selected a subset of the X-IIoTID dataset, containing 40,000 instances, each with 62 features, and the distribution of different classes is illustrated in Figure 5. During the dataset preprocessing phase, we removed columns named ‘is_SYN_with_RST’ and ‘Bad_checksum’ where all values were ‘False’. From columns labeled ‘class1’, ‘class2’, and ‘class3’ (‘class1’, ‘class2’, and ‘class3’, respectively, represent ‘normal and attack’, ‘normal and sub-category attack’, and ‘normal and sub-sub-category attack’.), we chose ‘class2’ as the label data. Subsequently, the data features were standardized. The processed dataset, after these procedures, took the shape of (40,000, 57). Finally, we employed the shuffle method to randomize the entire dataset and split it into training, validation, and testing sets in proportions of 80%, 10%, and 10%, respectively.

### 4.2. Parameter Settings of CTSF and Control Models

#### 4.2.1. Parameter Settings of CTSF

Prior to training CTSF, we reshaped the data to (40,000, 1, 57) to facilitate its passage through Conv1D for feature extraction.

The size and shape of the convolution kernel played a critical role in the CTSF, as the convolution operation involved sliding the kernel over the input data to extract features. This was essential because it directly affected how the model perceived and captured features from the input data. The CNN in the pre-training part of CTSF consisted of seven convolutional blocks, each comprising their own convolutional layer and max-pooling layer. The max-pooling layers in all seven blocks shared the same parameters (stride = 2, padding = 0). The size of the convolutional kernels within each convolutional block was set to 2, and the padding parameter was uniformly configured as 1. The parameter configurations for each convolutional block were as follows. in the first convolutional block, the convolutional layer had an input channel of 1 and an output channel of 2. In the second convolutional block, the convolutional layer featured 2 input channels and 4 output channels. Subsequently, the convolutional layers in the following blocks had an input channel count that was double that of the previous layer, with the output channel count also being double that of the previous layer.

The sizes of the convolutional layers mentioned above were all set to 2, which was highly advantageous for extracting local features from the input data, thereby maximizing and enhancing CTSF’s capability to extract local features effectively. If the size of the convolutional kernels is set larger than 2, theoretically, the local feature extraction capability of CTSF may not meet expectations. In the pre-training phase, to ensure that the input data could be passed through the CNN to the Transformer, the shape of the data coming out of the CNN had to have a third dimension of 1. Therefore, we configured the CNN part with 7 convolutional layers and 7 pooling layers. Each convolutional layer was accompanied by a max-pooling layer for feature reduction. After passing through the CNN section, the shape of the input data became (batch_size, 128, 1).

The pre-training section of CTSF’s Transformer consisted of a positional encoder, an encoder, an improved decoder, and a fully connected layer. The following are the parameter settings for each component:The parameter d_model in the positional encoder was set to 200, representing the extension of each feature of the input data to 200 dimensions. The dropout parameter was set to the default value of 0.1, and the max_len parameter limited the maximum number of features in the input data, which was set to the default value of 5000 in this study.The parameter d_model in the encoder was set to 200, consistent with the d_model parameter in the positional encoder; the nhead parameter was set to 2, indicating that there were 2 parallel heads in the multi-head attention layer to concurrently extract features from the input data; the nhid parameter was set to 2, indicating that the entire encoder consists of two encoder layers; the dropout parameter was set to 0.2 to ensure that the overall framework did not overfit as much as possible. The shape of the input data changed to (batch_size, 128, 200) after passing through the encoder section.The improved decoder consisted of a meticulously designed fully connected layer with an input feature dimension of 200 and an output feature dimension of 128, denoted as in_features and out_features, respectively. After passing through the decoder section, the shape of the input data changed to (batch_size, 128, 128).The section of fully connected layers consisted of a total of five layers. The parameters for the first fully connected layer: in_features with a value of 128 × 128 and out_features with a value of 1000. The second layer had 1000 as in_features and 500 as out_features. In the third layer, the in_features parameter was set to 500, and the out_features parameter was set to 200. For the fourth layer, in_features was 200 and out_features was 50. Finally, in the fifth layer, in_features was 50 and out_features was 10; here, 10 represents the number of classes for initialization classification. The activation functions employed were ReLU for the first four layers and softmax for the final layer.

The classification component of CTSF employed an SVM classifier for final categorization. Throughout the experimentation process, the kernel parameter was set to ‘rbf’, ‘poly’, ‘linear’, and ‘sigmoid’ for training. Section 4.4 demonstrates the classification performance of CTSF using various kernel functions.

#### 4.2.2. Parameter Settings of CNN-Transformer (Improved)

The parameters of CNN-Transformer (improved) were identical to those of CTSF’s pre-training phase. The reason for setting up this model as a control group was to demonstrate the effectiveness of the operation in CTSF where ample pre-training using the softmax function was performed on the data before passing the initialized classification data to the SVM.

#### 4.2.3. Parameter Settings of CNN-RNN

CNN-RNN comprises a CNN section, an RNN section, and a fully connected layer section. The parameters of the CNN section were the same as those of the CNN part in CTSF’s pre-training phase. After passing through the CNN section, the input data’s shape became (batch_size, 128, 1). The RNN section’s input_size parameter was set to 1, indicating that the input data had 128 features, each with a one-dimensional dimension. The hidden_size parameter was set to 150, representing 150 hidden layers in the RNN. The num_layers parameter was set to 2. The RNN section used the result of the last hidden layer and squeezed the output data using the squeeze function to make it two-dimensional. Finally, the input data’s shape became (batch_size, 150). The fully connected layer section comprised three fully connected layers. The in_features and out_features parameters of the first fully connected layer were set to 150 and 100, respectively. The parameters of the second fully connected layer were 100 and 50. The parameters of the third fully connected layer were 50 and 10.

#### 4.2.4. Parameter Settings of CNN

CNN comprises a CNN section and a fully connected layer section. The parameters of the CNN section were the same as those of the CNN part in CTSF’s pre-training phase. The fully connected layer section included three fully connected layers. The in_features and out_features parameters of the first fully connected layer were set to 128 and 100, respectively. The parameters of the second fully connected layer were 100 and 50. The parameters of the third fully connected layer were 50 and 10.

#### 4.2.5. Parameter Settings of RNN

RNN comprises a RNN section and a fully connected layer section. Before training the RNN model, we preprocessed the dataset into a shape of (40,000, 57, 1) to ensure smooth forward propagation in the RNN. The parameters of both components of this model remained consistent with those in the corresponding segments of the CNN-RNN model.

The training parameters of CTSF and the control group models are presented in Table 3.

### 4.3. Model Evaluation Metrics

The confusion matrix is a table commonly used in data science and machine learning to summarize the prediction results of classification models. It is represented by an n-by-n matrix, where records from the dataset are aggregated based on two criteria: ‘True Class’ and ‘Predicted Class’, with n representing the number of classes in the dataset. Taking binary classification as an example, the structure of the confusion matrix is illustrated in Table 4.

Accuracy is used to measure the prediction accuracy of a classification model, which is the ratio of correctly predicted samples to the total number of samples.


(19)
$$Accuracy=\frac{TP+TN}{TP+TN+FP+FN}$$


Precision measures the model’s ability to correctly predict positive instances among the predicted positives, i.e., the ratio of true positive instances to the total number of predicted positive samples.


(20)
$$Precision=\frac{TP}{TP+FP}$$


Recall measures the model’s ability to identify positive samples, i.e., the ratio of true positive instances to the total number of actual positive samples.


(21)
$$Recall=\frac{TP}{TP+FN}$$


F1 Score: an evaluation metric that comprehensively considers both precision and recall, calculated as the harmonic mean of precision and recall.


(22)
$$F1=\frac{2*Precision*Recall}{Precision+Recall}$$


### 4.4. Analysis and Discussion of Experimental Results

CTSF’s performance using different kernel functions is presented in Table 5, while the performance of the control group models is shown in Table 6.

The confusion matrices of CTSF using various kernel functions for classification are displayed in Figure 6. The confusion matrices of the other models in the comparative experiments are presented in Figure 7. In these matrices, ‘Exfiltration’, ‘Exploitation’, ‘Lateral_movement’, ‘Reconnaissance’, ‘Weaponization’, and ‘crvpto-ransomware’ are abbreviated as ‘Exf’, ‘Exp’, ‘L_m’ ‘Rec’, ‘Wea’, and ‘c-r’, respectively, for clarity.

In Table 5 and Table 6, the metrics Precision (PRE), Recall (RC), and F1-score (F1) are abbreviated. From Table 5, it can be concluded that the CTSF framework achieves the highest accuracy of 0.9885 when using the “rbf” kernel function and the lowest accuracy of 0.9365 when using the “sigmoid” kernel function. Figure 8 provides visualizations of CTSF’s performance using different kernel functions. The visualizations reveal that the decision boundary for the “Normal” class is less accurate when the “sigmoid” kernel function is used. In Table 4, the highest accuracy within the control groups is achieved by the CNN-Transformer (improved) model, with an accuracy of 0.9858. All three metrics related to the crypto-ransomware class reach 1, same as the results obtained by CTSF (with rbf); this indicates that the specialized design of the CTSF pretraining section equips the overall framework with a robust capability for detecting small-sample classes. Furthermore, the accuracy of the CNN-Transformer (improved) model is lower than that of CTSF using both the “rbf” and “linear” kernel functions. Since the parameters of CTSF’s pretraining section are identical to those of the control group’s CNN-Transformer model, this data corroborates the effectiveness of CTSF’s design (CTSF first pretrains the initialization classification results and subsequently passes them to SVM for final classification). Furthermore, from Table 5, we can observe that CTSF, using the ‘rbf’ kernel function, took 55,753.90 s to train for 500 epochs. Table 6 displays the training time for the models in the comparative experimental group, with the fastest training time achieved by the CNN model at 17,515 s. In fact, from the above data, it can be observed that while CTSF exhibits advantages in recognizing small sample classes and overall classification accuracy, it is not lightweight. We will delve into this limitation in more detail in Section 5.

Regarding the class “crvpto-ransomware”, which contains only 21 instances in this dataset, the CNN-RNN and CNN models exhibit poor performance, highlighting their limitations in handling small-sample data. On the contrary, the CTSF, CNN-Transformer (improved), and RNN models, employing non-”sigmoid” kernels, perform better with this class, demonstrating their efficacy in handling small-sample data.

Furthermore, Table 5 indicates that with identical CNN parameters, CNN-Transformer (improved) outperforms CNN-RNN in terms of classification accuracy. This suggests that our proposed improved version of Transformer excels in capturing global features compared to RNN.

Finally, Table 7 presents a comparison between CTSF and various benchmark models [40] on different attack class detection rates in X-IIoTID datasets. As per Table 7, CTSF achieved a detection rate of 1 for the “C&C” class, significantly higher than other benchmark models. Similarly, the “Weaponization” and “crypto-ransomware” classes also achieved a detection rate of 1. Additionally, while CTSF’s detection rate for some classes may be slightly lower than that of some models, its detection rates exhibit smoother fluctuations for all classes, avoiding scenarios where a high detection rate for one class is accompanied by a low detection rate for another class. Lastly, for the small-sample class “crypto-ransomware”, CTSF’s detection rate surpasses that of all benchmark models, indicating its advantage in accurately detecting such classes.

## 5. Conclusions and Future Work

In this paper, we propose a novel network framework, named CTSF, for Industrial Internet intrusion detection. The CNN in the pretraining part of CTSF is employed to extract local features from input data as comprehensively as possible. The Transformer in the pretraining part, tailored for Industrial Internet intrusion detection datasets, is an improved version that abandons the decoder’s self-attention mechanism due to the one-dimensional nature of label data in network traffic. This modification not only reduces the number of training parameters but also enhances the extraction of global features from input data. Ultimately, CTSF passes the well-pretrained data to SVM and selects an appropriate kernel function for classification.

To validate the efficacy of this framework in addressing intrusion-detection challenges in the context of Industrial Internet, we performed performance testing on a subset of X-IIoTID datasets, comparing it with advanced models as benchmarks. The experiments demonstrate that CTSF not only achieves a high accuracy but also accurately identifies suitable decision boundaries and correctly classifies instances, even in scenarios involving imbalanced or small sample data. Despite achieving high accuracy and excellent performance on small sample data, CTSF is not as lightweight in terms of computational resources. Heavier models typically require more computational resources to operate, which can potentially lead to higher hardware requirements, thereby increasing maintenance and operational costs. Furthermore, deploying less lightweight models into Industrial Internet systems may necessitate additional engineering resources, as they may require more extensive configuration and fine-tuning. Furthermore, the Industrial Internet environment also demands real-time capabilities, and the training and inference times of non-lightweight models can potentially become an issue. Therefore, looking ahead, we have the following considerations:In CTSF, local and global features of data are extracted using CNN and an enhanced Transformer. While this paper has improved and optimized the traditional Transformer decoder for the characteristics of Industrial Internet traffic data, making it more lightweight, CTSF as a whole is still not sufficiently lightweight. Next, we plan to explore optimization algorithms to further enhance the pre-training phase of CTSF, aiming to reduce its resource consumption while maintaining its accuracy and detection rate.In the pretraining phase, we can achieve parallel computing by adopting parallel training and parameter asynchronous updating methods. Distributing computational tasks to edge devices or within a distributed computing environment alleviates the burden on individual devices and reduces latency. Parallel and distributed computing can decompose the training tasks of large-scale models into multiple smaller tasks which are executed concurrently across various computing nodes, significantly reducing training time and enhancing training efficiency. It is worth noting that ensuring load balance among different nodes is a challenge in distributed computing. Some nodes may become busier than others, leading to performance imbalances. By taking into account both the benefits and challenges, and implementing appropriate optimization strategies, such as employing more efficient communication protocols and using suitable load balancing algorithms, the scalability and performance of the CTSF framework can be improved.Utilizing pruning algorithms to reduce the CTSF’s size. These algorithms can identify and remove connections or layers that contribute minimally to the CTSF’s accuracy.

## Figures and Tables

**Figure 1 sensors-23-08793-f001:**
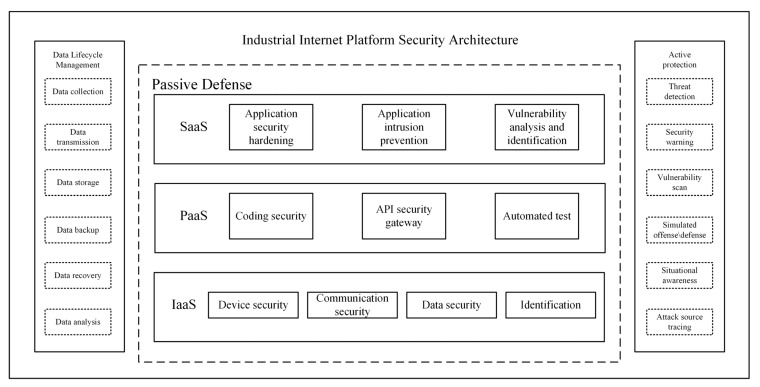
Industrial Internet platform security architecture.

**Figure 2 sensors-23-08793-f002:**
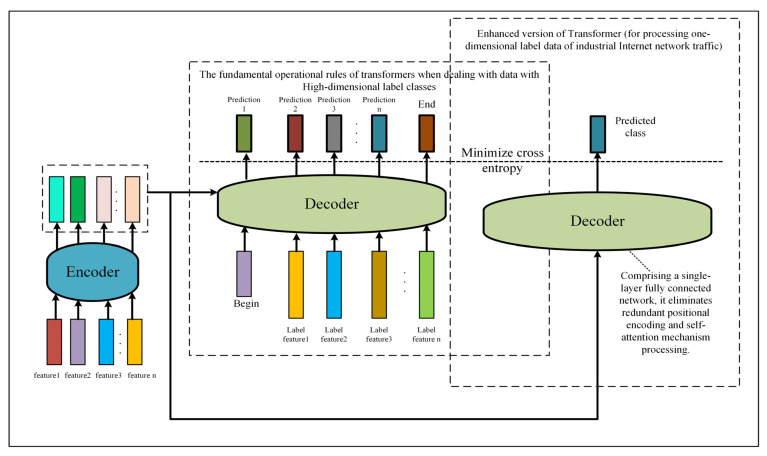
Comparison between the fundamental process of the Transformer in handling data with high-dimensional label classes and the basic process of the modified Transformer proposed in this paper for processing Industrial Internet network traffic data.

**Figure 3 sensors-23-08793-f003:**
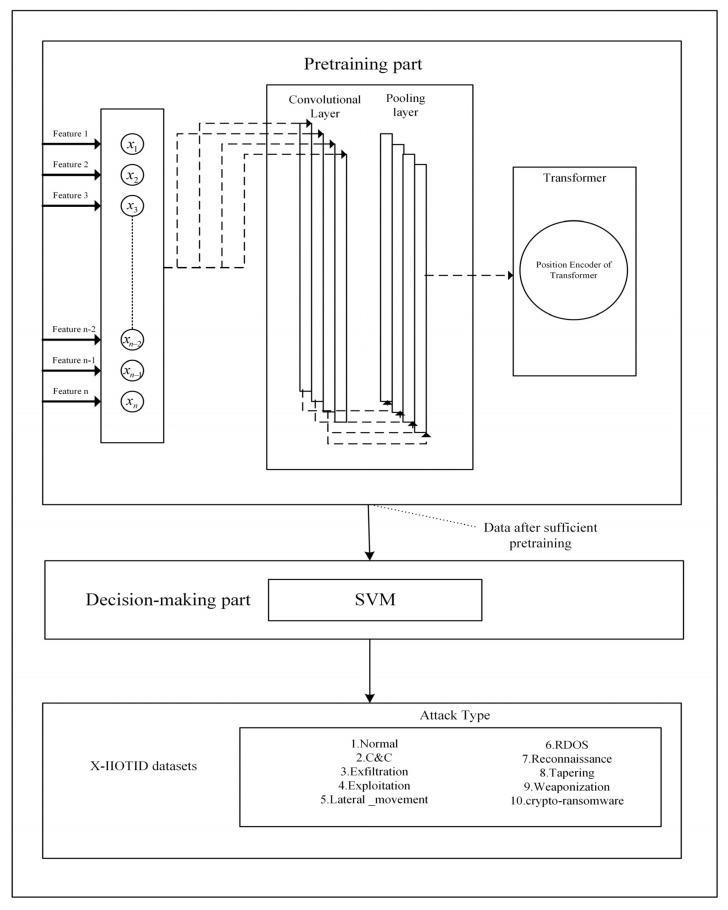
Basic structure of CTSF(The dotted arrows in the pre-training section represent the direction of data transfer from one module to another.).

**Figure 4 sensors-23-08793-f004:**
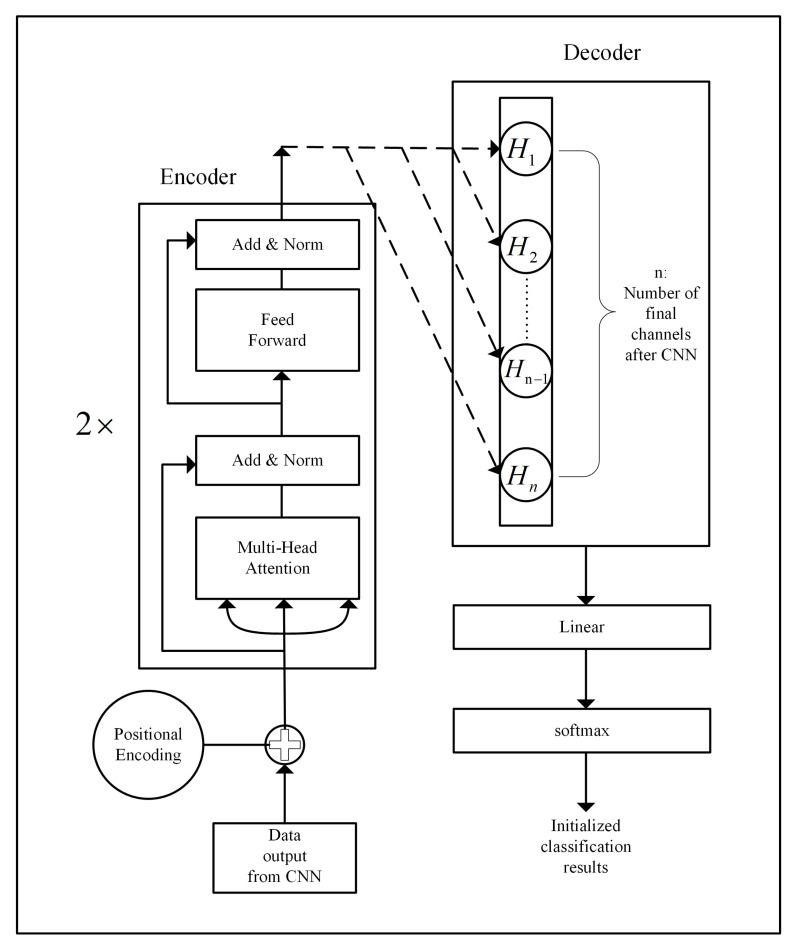
The improved Transformer architecture of the pre-training part. (The dotted arrows represent that not all data from the encoder part is input to each individual neuron in the enhanced decoder structure, but that the input of each feature corresponds one-to-one with the subscripts of *H*.)

**Figure 5 sensors-23-08793-f005:**
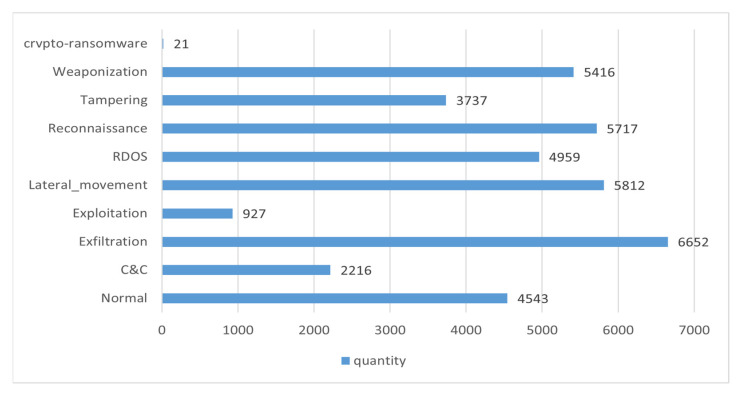
The number of various types in the data set selected in this paper.

**Figure 6 sensors-23-08793-f006:**
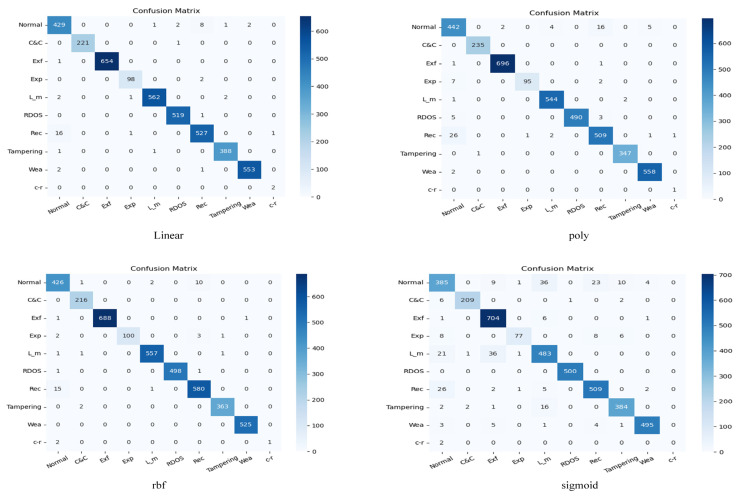
CTSF uses each kernel function to classify the confusion matrix.

**Figure 7 sensors-23-08793-f007:**
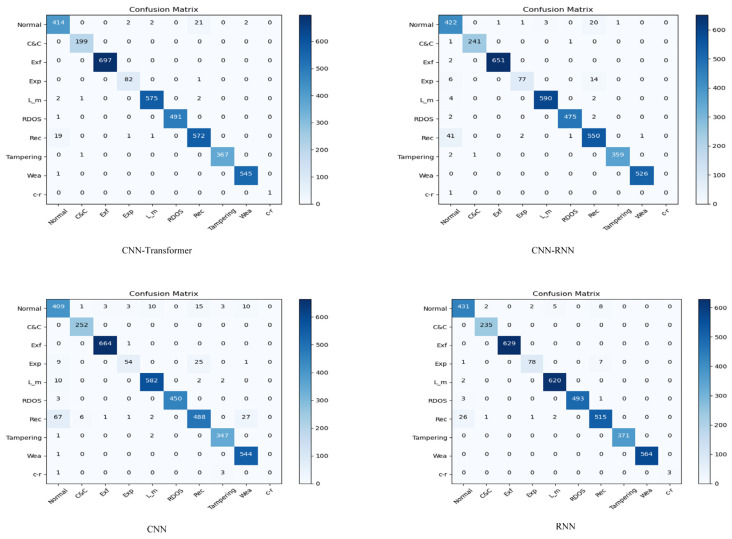
Confusion matrix of the model in the control experiment.

**Figure 8 sensors-23-08793-f008:**
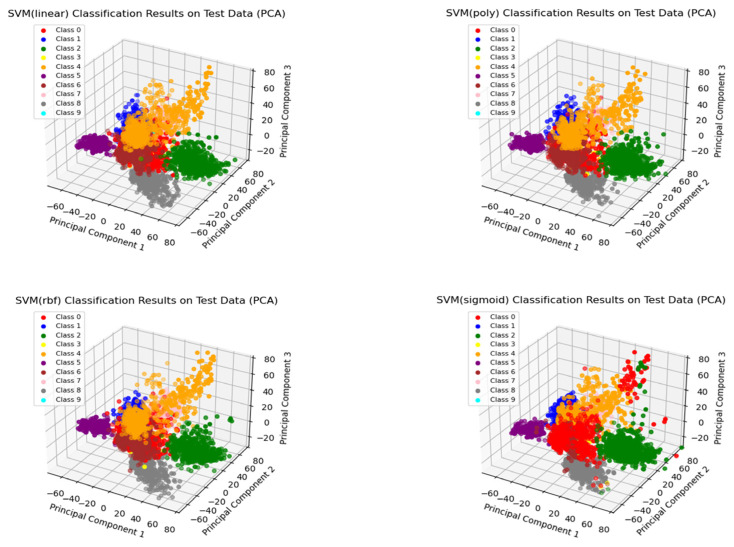
Visualization of the classification performance of CTSF with various kernel functions (Category numbers from 0 to 9 are: Normal, C and C, Exfiltration, Exploitation, Lateral_movement, RDOS, Reconnaissance, Tampering, Weaponization, crypto-ransomware).

**Table 1 sensors-23-08793-t001:** The parameters employed in the inference process of CTSF forward propagation and the corresponding meaning of each parameter.

No.	Parameters	Meaning
1	*n*	The number of features a piece of network traffic data has
2	X	A feature vector corresponding to a network traffic event
3	$N_{c}$	The “length” of the convolution kernel
4	$step_{c}$	The moving step of the convolution kernel
5	$X_{c\_i}$	The initial eigenvector of each step is convolved to obtain the process value
6	$w_{c}$	The weight parameters of the convolution kernel
7	$b_{c}$	The bias vector of the convolution kernel
8	$X_{c}$	The feature vector obtained after each convolution kernel of CTSF executes all steps
9	$N_{p}$	The “length” of $X_{c}$, it is numerically equal to n
10	$N_{m}$	The “length” of the max pooling kernel
11	$step_{m}$	The moving step size of the pooling kernel
12	$X_{m}$	The feature vector is obtained by $X_{c}$ through the maximum pooling layer
13	$X_{M}$	The feature vector obtained after X undergoes all convolution operations and pooling operations of CTSF
14	$X_{t}$	The feature vector obtained after X passes through the position encoder part of the CTSF
15	${\vec{p}}_{i}$	Represents the vector corresponding to position i, containing pairs of sine and cosine for each frequency.
16	a	Represents an integer value that controls the odd and even conditions in ${\vec{p}}_{i}$.
17	$\omega_{k}$	frequency
18	$f(i{)}^{\left(a\right)}:$	The function that generates the position vector ${{\vec{p}}_{i}}^{\left(a\right)}$
19	$d_{\mathrm{model}}$	Represents the dimension of each feature deepening
20	num	Numerically equal to the feature dimension of $X_{M}$
21	$W_{Q}$, $W_{K}$, $W_{V}$	Corresponds to the weight matrix of the query matrix, the weight matrix of the key matrix, and the weight matrix of the value matrix
22	$Q_{i}$	The query vector for the *i*-th header in the CTSF pre-training part
23	numh	The number of heads in the multi-head attention mechanism
24	$K_{i}$	The key vector for the *i*-th header in the CTSF pre-training section
25	$V_{i}$	The value vector for the *i*-th header in the CTSF pre-training section
26	$A_{i}$	The output of the self-attention mechanism of the *i*-th head
27	$X_{A}$	The results obtained by the self-attention mechanism of all heads are combined to obtain
28	μ,σ	represent the mean and standard deviation of $X_{A}$, respectively
29	$X_{NA}$	The vector obtained after $X_{A}$ passes through the standard layer
30	$F_{R1}$	Represents the process of mapping $X_{t}$ to $X_{NA}$
31	$X_{R1}$	CTSF establishes a residual connection at the multi-head attention mechanism layer, and this value represents the result of the residual connection
32	$F_{R2}$	Represents $X_{R1}$ through the mapping process of the fully connected layer and the normalized layer
33	$X_{R2}$	CTSF establishes a residual connection in the feedforward fully connected layer, and this value represents the result of the residual connection
34	$F_{l}$	Represents $X_{R2}$ through the mapping process of the fully connected layer
35	P	Represents the result of the original feature vector X after sufficient pre-training
36	$w_{f}$	The weight parameter matrix of the feed-forward fully connected layer of the CTSF pre-training part
37	$b_{f}$	The bias value vector of the feed-forward fully connected layer of the CTSF pre-training part
38	$w_{d}$	The weight parameter matrix of the decoder layer of the CTSF pre-training part
39	$b_{d}$	The bias value vector of the decoder layer of the CTSF pre-training part
40	$w_{l}$	Represents the weight parameters of the SVM
41	$b_{l}$	Represents the bias value of the SVM

**Table 2 sensors-23-08793-t002:** Pseudocode of CTSF overall training process.( The content after the ’#’ symbol is a comment.)

Input: train_data_iterator: It stores all the X and the attack types corresponding to X that are needed to train the model.
$w_{c},\;\;b_{c},\;\;W_{Q},W_{K},W_{V},w_{f},b_{f},w_{d},b_{d},w_{l},b_{l}:$ Matrix of weight parameters and bias values.
num_epochs: Total training epochs.
num_batches: The number of batches included in an epoch.
train_features: SVM training data (excluding categories), the type is an array.
train_labels: Corresponding to train_features, representing the category of training data, the type is also an array.
Process:
preprocess_data(X) # Preprocess the dataset.
initialize_parameters ($w_{c},b_{c},W_{Q},W_{K},W_{V},w_{f},b_{f},w_{d},b_{d},w_{l},b_{l}$) # Initialize weight parameters.
for epoch in range(num_epochs):
total_loss = 0
for batch_data, batch_labels in train_data_iterator:
predictions = pretraining_CTSF (batch_data)
loss = compute_loss(predictions, batch_labels)
total_loss += loss
backpropagation(loss)
update_parameters($w_{c},b_{c},W_{Q},W_{K},W_{V},w_{f},b_{f},w_{d},b_{d}$)
end for
average_loss = total_loss/num_batches
end for
for batch_data, batch_labels in train_data_iterator:
batch_features = pretraining_CTSF.extract_features(batch_data) # Obtain data from the pre-training part.
train_features.append(batch_features)
train_labels.append(batch_labels)
end for
svm_CTSF (train_features, train_labels) # update${\;w}_{l}$ and $b_{l}$
Output:
predictions = svm_CTSF.predict(test_features)

**Table 3 sensors-23-08793-t003:** Training parameters for CTSF and control models.

Model	Batch Size	Optimizer	Epochs	Learning Rate	Gamma	Step_Size	SVM C	SVM Degree
CTSF	32	Adadelta	500	0.01	0.65	50	1	3
CNN-Transformer (improved)	32	Adadelta	500	0.01	0.65	50	NULL	NULL
CNN-RNN	32	Adadelta	500	0.01	0.65	50	NULL	NULL
CNN	32	Adadelta	500	0.01	0.65	50	NULL	NULL
RNN	32	Adadelta	500	0.01	0.65	50	NULL	NULL

**Table 4 sensors-23-08793-t004:** Confusion matrix.

Total (P + N)		Predicted Condition	
		Postive (PP)	Negative (PN)
Actual condition	Postive (P)	True positive (TP)	False negative (FN)
	Negative (N)	False positive (FP)	True negative (TN)

**Table 5 sensors-23-08793-t005:** Performance of CTSF using different kernel functions.

	Linear			rbf			Sigmoid			Poly		
	PRE	RC	F1	PRE	RC	F1	PRE	RC	F1	PRE	RC	F1
Normal	0.951	0.968	0.959	0.950	0.970	0.960	0.848	0.822	0.835	0.913	0.942	0.927
C and C	1	0.995	0.997	0.981	1	0.990	0.985	0.958	0.972	0.995	1	0.997
Exfiltration	1	0.998	0.999	1	0.997	0.998	0.929	0.988	0.958	0.997	0.997	0.997
Exploitation	0.980	0.980	0.980	1	0.943	0.970	0.962	0.777	0.860	0.989	0.913	0.95
Lateral_movement	0.996	0.991	0.993	0.994	0.994	0.994	0.882	0.891	0.887	0.989	0.994	0.991
RDOS	0.994	0.998	0.996	1	0.996	0.997	0.998	1	0.999	1	0.983	0.991
Reconnaissance	0.977	0.966	0.972	0.976	0.973	0.974	0.935	0.933	0.934	0.958	0.942	0.950
Tampering	0.992	0.994	0.993	0.994	0.994	0.994	0.952	0.948	0.950	0.994	0.997	0.995
Weaponization	0.996	0.994	0.995	0.998	1	0.999	0.986	0.972	0.979	0.989	0.996	0.992
crypto-ransomware	0.667	1	0.800	1	1	1	0	0	0	0.5	1	0.667
macro avg	0.960	0.990	0.970	0.990	0.990	0.99	0.85	0.83	0.84	0.93	0.98	0.95
weighted avg	0.990	0.990	0.990	0.990	0.990	0.99	0.93	0.93	0.93	0.98	0.98	0.98
Accuracy	0.98825			0.9885			0.9365			0.97925		
Training Time(s)	55,755.99			55,753.90			55,752.77			55,751.52		

**Table 6 sensors-23-08793-t006:** Performance of the control model.

	CNN-Transformer (Improved)			CNN-RNN			CNN			RNN		
	PRE	RC	F1	PRE	RC	F1	PRE	RC	F1	PRE	RC	F1
Normal	0.947	0.938	0.943	0.877	0.941	0.908	0.816	0.901	0.856	0.930	0.962	0.946
C and C	0.990	1	0.995	0.995	0.991	0.993	0.972	1	0.986	0.987	1	0.993
Exfiltration	1	1	1	0.998	0.996	0.997	0.994	0.998	0.996	1	1	1
Exploitation	0.964	0.987	0.976	0.962	0.793	0.870	0.915	0.606	0.729	0.962	0.906	0.934
Lateral_movement	0.994	0.991	0.993	0.994	0.989	0.992	0.976	0.976	0.976	0.988	0.996	0.992
RDOS	1	0.997	0.998	0.995	0.991	0.993	1	0.993	0.996	1	0.991	0.995
Reconnaissance	0.959	0.964	0.962	0.935	0.924	0.929	0.920	0.824	0.869	0.969	0.944	0.957
Tampering	1	0.997	0.998	0.997	0.991	0.994	0.977	0.991	0.984	1	1	1
Weaponization	0.996	0.998	0.997	0.998	1	0.999	0.934	0.998	0.965	1	1	1
crypto-ransomware	1	1	1	0	0	0	0	0	0	1	1	1
macro avg	0.99	0.99	0.99	0.88	0.86	0.87	0.85	0.83	0.84	0.98	0.98	0.98
weighted avg	0.99	0.99	0.99	0.97	0.97	0.97	0.95	0.95	0.95	0.98	0.98	0.98
Accuracy	0.9858			0.9728			0.9475			0.9847		
Training Time(s)	55,750			25,805			17,515			17,915		

**Table 7 sensors-23-08793-t007:** Comparison results of various benchmark models and CTSF in different attack class detection rates for X-IIoTID datasets.

Model	C and C	Exfiltration	Exploitation	Lateral_Movement	RDOS	Reconnaissance	Tampering	Weaponization	Crypto-Ransomware
DT	0.8966	0.8976	0.9852	0.9948	0.9999	0.9922	0.9947	0.9997	0.9986
NB	1	0.9823	0.5267	0.0834	0.9906	0.0150	0.9912	0.9867	0.9962
KNN	0.8096	0.7140	0.9128	0.9862	0.9999	0.9688	0.7868	0.9985	0.9960
SVM	0.8194	0.8394	0.8987	0.9983	0.9996	0.9170	0.9891	0.9996	0.9992
LR	0.5801	0.4634	0.7842	0.9809	0.9983	0.8666	0.7407	0.9907	0.9986
DNN	0.7716	0.9991	0.8129	0.9769	0.9996	0.9585	0.9815	0.9994	0.8188
GRU	0.8826	0.9993	0.8623	0.9789	0.9998	0.9886	0.9918	0.9997	0.9694
CTSF (with rbf)	1	0.997	0.943	0.994	0.996	0.973	0.994	1	1

## Data Availability

The original data can be obtained by contacting the corresponding author.

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
