# Peer review of "CTSF: An Intrusion Detection Framework for Industrial Internet Based on Enhanced Feature Extraction and Decision Optimization Approach"

_sensors, 2023, doi:10.3390/s23218793_

Round 1

Reviewer 1 Report

Comments and Suggestions for Authors

Summary: The main contribution of the work is the development of a novel network framework, CTSF (Convolutional-Transformer-SVM Framework), for intrusion detection in the industrial internet context. CTSF combines a Convolutional Neural Network (CNN) and an enhanced version of the Transformer model to capture both local and global features of input data while reducing feature dimensions. The framework incorporates Support Vector Machines (SVM) in the decision-making component to identify optimal decision boundaries. Experimental results show that CTSF achieves high accuracy and effectively discriminates minor classes, demonstrating its superiority in intrusion detection tasks for industrial internet data.

Suggestions and Comments:

  1. Clarify the specific goals and objectives of the CTSF framework in the abstract to provide a clearer understanding of its purpose.
  2. Provide more details about the specific types of devices, sensors, and control systems that generate the network traffic data in the industrial internet context.
  3. Elaborate on the challenges and limitations faced by traditional intrusion detection frameworks, highlighting the specific areas where the CTSF framework aims to overcome those challenges.
  4. Provide a high-level overview of the pre-training section and decision-making section of the CTSF framework in the abstract, giving readers a better understanding of the framework's overall structure.
  5. Explain the significance and relevance of the parameters presented in Table 1, providing more context on how they contribute to the forward propagation process of CTSF.
  6. Provide more information about the convolutional kernel used in the pre-training phase, including its size, shape, and how it relates to the features of the input data.
  7. Clarify the purpose and role of the encoder and decoder sections in the pre-training phase's Transformer structure, explaining how they contribute to feature extraction and dimensionality reduction.
  8. Provide a clear explanation of the decision-making part, discussing how the retraining process with SVM enhances the framework's generalization capability and improves classification accuracy.
  9. Include specific details about the X-IIoTID dataset used for evaluating the CTSF framework, such as the size, composition, and characteristics of the dataset.
  10. Provide a more detailed comparison of the CTSF framework with the CNN, RNN, CNN-RNN, and CNN-Transformer models, highlighting the specific metrics used and the performance differences observed.
  11. Include a discussion of the computational resource requirements of the CTSF framework, addressing potential challenges and considerations in deploying the framework in industrial Internet systems.
  12. Elaborate on the optimization algorithms that could be explored to reduce the resource consumption of the CTSF framework while maintaining its accuracy and detection rate in the future work section.
  13. Discuss the potential benefits and challenges of parallel computing and distributing computational tasks in the context of the CTSF framework, providing more details on how these approaches can enhance scalability and system performance.
  14. The authors are invited to include some recent references, especially those related to  Deep Convolutional Neural Networks.
  15.   For instance, the authors may include the following interesting references (and others):

    a. https://www.mdpi.com/2073-431X/12/8/151

    b. https://www.taylorfrancis.com/chapters/edit/10.1201/9781003393030-10/learning-modeling-technique-convolution-neural-networks-online-education-fahad-alahmari-arshi-naim-hamed-alqa

Comments on the Quality of English Language

Can be improved

Reviewer 2 Report

Comments and Suggestions for Authors

The paper proposes an IDS for IIoT data, combining CNNs. Transformers and SVM for traffic classification. The proposed solution has a high overall accuracy (98.875%) and performs well also for "small" classes.
The main drawback of the proposed solution is its computational complexity, which is mentioned only in the conclusions without experimental data. The comparison with other methods should include the computation time (both for training and testing), discussing the applicability for on-line detection.
Another weakness is related to the choice of the architectural parameters of the framework - it would be nice to provide some insights on the different choices (as done, for instance, for the different kernel functions) at least for some of them.
Moreover, the models considered in table 7 should be a little bit explained, at least adding a reference as done in [40] - however it seems that the data in the paper are not taken from the reference since a wider dataset is considered there.  

Finally, a couple of minor issues:
- the definition of confusion matrix (see table 4) is not clear
- in section 3 the symbols used in the formulas should be used also in the figures

Comments on the Quality of English Language

In general, the paper is well written and easy to read also for non experts. I suggest the following minor changes:
- Consider revising the text on page 8, where there are several separated subordinate clauses

- lines 402-418 could be written in more compact form 

- the seme is true for the definitions of TP, TN, FP and FN (probably they can be omitted)

Reviewer 3 Report

Comments and Suggestions for Authors

This paper introduces the CTSF network framework designed for detecting intrusions in the industrial internet. The framework is comprised of two main components: a pre-training component that incorporates Convolutional Neural Networks (CNN) and an enhanced Transformer, and a decision-making component that utilizes Support Vector Machines (SVM). The proposed framework was evaluated on the X-IIOTID dataset to demonstrate its effectiveness.

This paper presents an interesting approach to industrial internet intrusion detection. Overall, it is well-structured and clearly presented. However, there are some areas that need improvement, and some points of clarification:

- The paper mentions that the framework captures both local and global features from input data. It would be beneficial to explain why capturing local features is crucial for industrial internet intrusion detection when it might not be as essential for other applications. Are there any empirical studies or heuristics that support this claim?

- Figure 1 illustrates a generalized security architecture for safeguarding Industrial Internet platforms but lacks explanations. Providing explanations of the content in both Figure 1 and Figure 2 would enhance the reader's understanding.

- On line 131-136, ‘Chapter’ should be replaced with ‘Section’ to make it Section 2, Section3, etc.

- In Section 2.3, it would be beneficial to briefly describe the differences or advantages of the proposed model compared to those presented in references [34 - 36] since they are also intrusion detection systems.

- There is no definition of m in formula (3). m is not included in Table 1.

- There is no definition of (a) in position vector in formula (8), whereas this is not present in formula (10).

- In Section 4.1, it would be valuable to provide explanations or sample data for 'class1', 'class2', and 'class3'.

- The benchmark models listed in Table 7 lack references. Considering the critical aspect of prompt response in intrusion detection, does the proposed model exhibit superior or similar time efficiency compared to the benchmark comparison models? Additionally, the comparison appears to be somewhat limited in scope. Table 7 includes seven benchmark comparison models, while reference [40] has nine benchmark comparison models. Furthermore, it is not clear whether the proposed model was compared to the models in [34-36, 40].

Reviewer 4 Report

Comments and Suggestions for Authors

The article is relevant. It directly addresses the issues of information security. The authors analyze attempts at Internet attacks and ways to prevent them based on economic and mathematical modeling models. The article has a pronounced practical aspect, is distinguished by scientific novelty and can be recommended for publication.

Recommendations for the revision of the article:

1. Add the purpose and objectives of the article to the abstract.

2. Form a research hypothesis and prove it in the conclusion of the article.

3. It is not clear whether the drawings 1,2,3 are copyrighted or not? Give a link to the source or indicate that they are copyrighted

4. There are no units of measurement in table 1

5. Check the entry in formula 1, the interval from 0 to 0 is specified.

6. In Figure 5, remove the legend and the name of the figure from above, because there is already a similar inscription at the bottom

Round 2

Reviewer 1 Report

Comments and Suggestions for Authors

The authors considered my comments and suggestions 

Comments on the Quality of English Language

Can be improved 

Reviewer 2 Report

Comments and Suggestions for Authors

The authors addressed most of the required changes, so the paper can be accepted in the present form

Reviewer 3 Report

Comments and Suggestions for Authors

The authors have taken my suggestions, and address / clearly explain my concerns in the response to my comments. I have no further questions.